# The Absence of Adiponectin Alters Niacin’s Effects on Adipose Tissue Inflammation in Mice

**DOI:** 10.3390/nu12082427

**Published:** 2020-08-13

**Authors:** Emily C. Graff, Han Fang, Desiree Wanders, Robert L. Judd

**Affiliations:** 1Department of Pathobiology, Auburn University, Auburn, AL 36849, USA; ecg0001@auburn.edu; 2Scott-Ritchey Research Center, Auburn University, Auburn, AL 36849, USA; 3Pennington Biomedical Research Center, Louisiana State University, Baton Rouge, LA 70808, USA; han.fang@pbrc.edu; 4Department of Nutrition, Georgia State University, Atlanta, GA 30302, USA; dwanders@gsu.edu; 5Department of Anatomy, Physiology and Pharmacology, Auburn University, Auburn, AL 36849, USA

**Keywords:** obesity, inflammation, immune, niacin, adipokines

## Abstract

Obesity is an immunometabolic disease associated with chronic inflammation and the dysregulation of pro- and anti-inflammatory cytokines. One hallmark of obesity is reduced concentrations of the anti-inflammatory adipokine, adiponectin. Pharmacologic doses of niacin produce multiple metabolic benefits, including attenuating high-fat diet (HFD)-induced adipose tissue inflammation and increasing adiponectin concentrations. To determine if adiponectin mediates the anti-inflammatory effects of niacin, male C57BL/6J (WT) and adiponectin null (*Adipoq*^-/-^) mice were maintained on a low-fat diet (LFD) or HFD for 6 weeks, before being administered either vehicle or niacin (360 mg/kg/day) for 5 weeks. HFD-fed mice had increased expression of genes associated with macrophage recruitment (*Ccl2*) and number (*Cd68*), and increased crown-like structure (CLS) number in adipose tissue. While niacin attenuated *Ccl2* expression, there were no effects on *Cd68* or CLS number. The absence of adiponectin did not hinder the ability of niacin to reduce *Ccl2* expression. HFD feeding increased gene expression of inflammatory markers in the adipose tissue of WT and *Adipoq^-/-^* mice. While niacin tended to decrease the expression of inflammatory markers in WT mice, niacin increased their expression in HFD-fed *Adipoq^-/-^* mice. Therefore, our results indicate that the absence of adiponectin alters the effects of niacin on markers of adipose tissue inflammation in HFD-fed mice, suggesting that the effects of niacin on tissue cytokines may involve adiponectin.

## 1. Introduction

Obesity is a chronic low-grade inflammatory disease characterized by increased inflammatory cytokines, circulating immune cells, and immune cells within adipose tissue [1,2,3,4,5,6]. Specific pathophysiologic features of human and rodent models of obesity include increased numbers of adipose tissue macrophages (ATMs) and a shift in macrophage polarity from an alternatively activated (M2) to a classically activated (M1) state [7]. Clinically, obesity and chronic adipose tissue inflammation are associated with the development of metabolic syndrome, insulin resistance, type 2 diabetes, and low-grade systemic inflammation [8,9,10]. In humans, chronic obesity and adipose tissue inflammation are associated with increases in serum MCP-1, TNFα, IL-6 [8,11,12,13], and C-reactive protein (CRP) [14], and the increased circulation of pro-inflammatory mononuclear cells [1,15].

Along with inflammation, adipokine dysregulation—specifically decreased adiponectin [16,17,18,19] and increased leptin [20]—is a hallmark of obesity in humans and in rodent models of obesity [16,17,18,19,20]. Decreases in adiponectin precede the development of insulin resistance [18]. Adiponectin concentrations are inversely correlated to adipose tissue mass, and decreased serum adiponectin is associated with the development of metabolic disease, atherosclerosis, and type 2 diabetes [17,18,19]. Adiponectin is an anti-inflammatory mediator that plays a role in ATM polarization, immune cell migration, and chemokine production [21,22,23,24]. In cell culture, adiponectin inhibits the phagocytic activity of mature human macrophages [25] and in vivo studies indicate that peritoneal macrophages from adiponectin null mice exhibit an increased M1 profile [22]. Others have shown that treatment with adiponectin decreases TLR-4 receptors on cultured macrophages and shifts macrophages towards an M2 phenotype [23]. Overall, maintenance of an adequate adiponectin concentration appears to be tied to improved adipocyte function, metabolic homeostasis, and decreased inflammation [21].

For over 50 years, pharmacologic doses of niacin have been used in the treatment of atherogenic dyslipidemia [26], which is associated with obesity and metabolic syndrome. However, recent evidence indicates that the beneficial effects of niacin may involve the modulation of inflammatory pathways that are independent of its anti-lipolytic effects on adipocytes [27,28,29]. Beneficial effects of niacin have been reported in adipocytes, immune cells, and vascular endothelial cells [28,29,30,31,32,33]. Both macrophages and mature adipocytes express the receptor for niacin, HCA_2_ [34,35,36]. Activation of the HCA_2_ receptor has anti-inflammatory effects throughout numerous organ systems, including the gastrointestinal, cardiovascular, and nervous systems [37]. In 2011, Lukasova et al. showed that niacin-mediated HCA_2_ activation decreases M1 macrophage differentiation in atherosclerotic plaques in mice [29]. Similar findings are observed in adipose tissue of mice, as the chronic administration of niacin attenuates the HFD-induced expression of M1 macrophage markers and chemokines such as MCP-1 in an HCA_2_-dependent manner [28]. In addition, treatment with niacin has been shown to inhibit macrophage recruitment, adhesion, and chemotaxis, and to alter TLR-mediated cytokine release [26,27,30,38]. In vascular endothelial cells, niacin inhibits acute vascular inflammation, oxidative stress, and monocyte adhesion [31,32,33]. Previous work from our laboratory and others demonstrates that niacin increases serum concentrations of adiponectin in rodent models and humans [30,39,40,41], and this increase is mediated by the HCA_2_ receptor [28,42]. Therefore, niacin may decrease inflammation either through direct effects on adipocytes, macrophages and endothelial cells, or indirectly, by altering systemic pro- and anti-inflammatory factors such as adiponectin. Likewise, niacin and adiponectin may act synergistically to decrease inflammation associated with obesity.

It is not known if the anti-inflammatory effects of niacin are mediated by adiponectin. Thus, the objective of this study is to determine whether adiponectin is necessary for niacin to decrease HFD-induced adipose tissue inflammation. Using wild-type (WT) and *Adipoq*^-/-^ mice, we show that niacin decreased HFD-induced body weight gain in an adiponectin-dependent manner, but the effects of niacin on inflammatory markers differed between genotypes. Niacin tended to decrease inflammatory markers in WT mice but increased them in *Adipoq*^-/-^ mice. One unexpected finding was that niacin significantly increased *Arg1* (a classic M2 macrophage marker) in *Adipoq*^-/-^ mice. These results indicate a complex relationship between adiponectin and the mechanisms by which niacin exerts its anti-inflammatory effects.

## 2. Materials and Methods

### 2.1. Animal Studies

All animal studies were approved by the Auburn University Institutional Animal Care and Use Committee prior to initiation, and all methods were performed in accordance with relevant guidelines. Thirty-two 3-week-old male C57BL/6J mice were purchased from Jackson Laboratories (Bar Harbor, ME, USA). Mice were maintained on either a low-fat diet (LFD; 10% kcal as fat; *n* = 16) or a high-fat diet (HFD; 60% kcal as fat; *n* = 16) obtained from Research Diets (New Brunswick, NJ, USA) for 11 weeks. After six weeks on the low-fat or high-fat diets, half of the mice received niacin (Sigma-Aldrich, St. Louis, MO, USA) at approximately 360 mg/kg/day dissolved in drinking water or vehicle (water) for five weeks. Water intake was measured every 4 days and niacin concentration in the water was adjusted based on water consumption. After five weeks of vehicle or niacin treatments, mice were fasted overnight (12 h) and euthanized by decapitation. Whole blood was collected, and serum was isolated. Tissues were flash frozen in liquid nitrogen and stored at −80 °C until analysis. Parallel studies were conducted in male global adiponectin null mice derived from a breeding pair of B6.129-*Adipoq^tm1Chan^*/J (*Adipoq^-/-^*) mice purchased from Jackson Labs. Thirteen *Adipoq^-/-^* mice were placed on the LFD diet (6 received vehicle; 7 received niacin), and sixteen *Adipoq^-/-^* mice were placed on the HFD (8 received vehicle; 8 received niacin).

### 2.2. Serum Analysis

Serum total adiponectin concentrations were measured by ELISA (Millipore; Temecula, CA, USA). Serum triglyceride concentrations were measured using a colorimetric assay (Cayman Chemicals; Ann Arbor, MI, USA). Blood glucose concentrations were determined using the AccuChek Active Handheld glucometer (Roche, Indianapolis, IN, USA) on whole blood collected from the mandibular vein. Serum NEFAs were measured using a colorimetric assay (Wako Chemicals; Richmond, VA, USA). Serum insulin, IL-6, MCP-1, and TNF-α were measured using Milliplex multiplex custom kit (Millipore; Temecula, CA, USA) and read on a MAGPix Bio-analyzer (Luminex; Temecula, CA, USA).

### 2.3. Gene Expression Analysis

RNA was isolated from epididymal white adipose tissue (F) using Qiagen RNeasy Lipid Tissue Mini Kit with on-column DNA digestion (Qiagen, Valencia, CA, USA). RNA (0.5 or 1 µg) was reverse transcribed into cDNA using an iScript cDNA Synthesis Kit from Bio-Rad (Bio-Rad, Hercules, CA, USA). PCR primers used in the real-time PCR analysis were previously published [28]. Analyses were performed on a Bio-Rad iCycler iQ thermocycler. Samples were analyzed in 30 µL reactions using SYBR Green PCR Master Mix (Bio-Rad, Hercules, CA, USA). All expression levels were normalized to the corresponding *Rppl0* mRNA levels, and analyzed using the 2^−ΔΔCT^ method [43]. *Rppl0* mRNA levels were unchanged in response to HFD or niacin treatment.

### 2.4. Immunoblot Analysis

EWAT pads (~100 mg) were homogenized with a handheld homogenizer in Pierce RIPA buffer (Rockford, IL, USA) supplemented with cOmplete® protease inhibitor tablet (Roche, Indianapolis, IN, USA) and phosphatase inhibitors (Sigma-Aldrich, St. Louis, MO, USA). The protein fractions were isolated and a DC protein assay (Bio-Rad, Hercules, CA, USA) was conducted to determine the protein concentration for each sample. Proteins (20 µg) were boiled for 20 min in 5% 2-mercaptoethanol to allow for the assessment of fully denatured and reduced monomers of adiponectin, which were then separated by SDS–PAGE (10%) and transferred to nitrocellulose membranes. Membranes were blocked in blocking buffer (LI-COR, Lincoln, NE, USA) for 1 hour and incubated with primary antibody overnight at 4 °C. Primary antibodies were either a rabbit polyclonal adiponectin antibody acquired from Abcam, Inc. (Cambridge, MA, USA) or a GAPDH mouse monoclonal antibody from Millipore (Billerica, MA, USA). The membranes were then washed with PBS with 0.1% Tween-20 and incubated with secondary antibody (either goat anti-mouse IRDye 800CW or donkey anti-rabbit IRDye 680RD (LI-COR, Lincoln, NE, USA)), and washed with PBS with 0.1% Tween-20. Blots were imaged using the LI-COR Odyssey scanner (LI-COR, Lincoln, NE, USA) with band density quantified using Image Studio software ver. 2.0 (LI-COR, Lincoln, NE, USA).

### 2.5. Histopathologic Analysis of EWAT

Portions of the EWAT were immediately fixed in 10% neutral buffered formalin, processed overnight and paraffin-embedded. Five-micron-thick hematoxylin and eosin (H&E)-stained sections were used for evaluation, and macrophage identification was verified with F4/80 cytoplasmic staining in step sections using rat anti-mouse F4/80 antibody (Abd Serotec, Raleigh, North Carolina; Clone CI:A3-1) with Rodent Block M (BioCare Medical, Concord, CA, USA) to reduce nonspecific staining. Crown-like structures (CLS) were defined as shrunken adipocytes surrounded by morphologically identified macrophages [6]. Slides were scanned using the Aperio ScanScope scanner (Vista, CA, USA) and evaluated on VisioPharm software (Hoersholm, Denmark) as previously described [44]. Briefly, the entire adipocyte section was evaluated to identify individual adipocytes using the following rules. Objects were identified and counted as adipocytes if: 1) the area was between 500 and 20,000 µm; and 2) it had a shape factor of 0–0.7, where a shape factor of 1 indicates a straight line and 0 indicates a perfect circle. Approximately 80–90% of the adipocytes were counted for each section. CLS were counted on H&E-stained slides by a board-certified pathologist and the entire surface area of each fat pad was counted to provide a total number of CLS per fat pad, which was then normalized per 100 adipocytes based on the total adipocyte number per section.

### 2.6. Statistical Analysis

All data sets were evaluated for normality and data that did not exhibit Gaussian distribution were log transformed. Analyses of serum and tissue protein concentrations and relative gene expression were determined by two-way ANOVA with Tukey’s post-hoc analysis. All statistical analyses were performed on Graph Pad Prism 7 software (La Jolla, CA, USA).

## 3. Results

### 3.1. Effects of HFD and Niacin on Metabolic Parameters in Mice

At the start of the study there was no statistical difference in body weight observed in any of the treatment groups. At the end of the study, as expected, compared to LFD-fed mice, HFD-fed mice had significantly increased body weight, EWAT weight, and blood glucose concentrations (Figure 1). Neither niacin treatment nor genotype affected any of these parameters. Over the course of the study, the HFD-fed mice consumed more calories than LFD-fed mice, but neither genotype nor niacin had any effect on food or water intake (Figure 1C). HFD feeding increased serum insulin concentrations and HOMA-IR, with these effects being more pronounced in *Adipoq*^-/-^ mice (Figure 1E,F).

### 3.2. Effects of HFD and Niacin on Adiponectin

HFD produced a ~2-fold decrease in adiponectin gene expression (Figure 2A) and a trend towards decreased adiponectin protein expression in adipose tissue (Figure 2B). In LFD-fed mice, niacin had no effect on adipose tissue adiponectin mRNA (Figure 2A) or total protein expression within the tissue (Figure 2B). However, niacin treatment significantly increased adiponectin mRNA and protein expression in HFD-fed mice (Figure 2A,B). Interestingly, niacin significantly increased serum adiponectin concentrations by ~14% in LFD-fed mice but had no effect on serum adiponectin concentrations in HFD-fed mice (Figure 2C). *Adipoq^-/-^* mice did not have measurable adiponectin protein concentrations in EWAT (Figure 1B).

### 3.3. Effects of HFD and Niacin on Markers of Macrophage Recruitment and Number

HFD feeding increased gene expression of the general macrophage marker *Cd68* and the pro-inflammatory chemokine *Ccl2* in adipose tissue of WT and *Adipoq*^-/-^ mice (Figure 3A,B). 

Interestingly, HFD-fed *Adipoq^-/-^* mice had significantly greater *Ccl2* gene expression than WT HFD-fed mice. Treatment with niacin had no effect on *Cd68* gene expression in any group. However, niacin attenuated the HFD-induced increase in *Ccl2* gene expression in *Adipoq^-/-^* mice. While it did not reach statistical significance, niacin reduced *Ccl2* gene expression two-fold in WT HFD-fed mice. In agreement with the HFD-induced increase in inflammatory gene expression, HFD-fed mice also had a significant increase in the number of CLS in EWAT, as well as in adipocyte size (Figure 4). The HFD-induced increase in CLS number in *Adipoq^-/-^* mice did not reach statistical significance, likely due to the low number of CLS present overall. Niacin treatment had no effect on the presence of CLS or on adipocyte size in either genotype (Figure 4C,D).

### 3.4. Effects of HFD and Niacin on Other Markers of Adipose Tissue Inflammation

In general, HFD increased gene expression of pro-inflammatory markers in adipose tissue of WT and *Adipoq^-/-^* mice (Figure 5A,B), with a significant increase noted in *Itgax* gene expression in WT mice, and a significant increase in *Tnf* gene expression in *Adipoq^-/-^* mice. Interestingly, niacin tended to reduce *Itgax* expression in WT mice (*p* = 0.07), but significantly increased *Itgax* expression in *Adipoq^-/-^* mice. Niacin treatment had no significant effect on *Tnf* expression in EWAT.

As part of general adipose tissue inflammation, additional markers including *Arg1* and *Mrc1* were evaluated. HFD had no effect on *Arg1* expression in WT mice, but significantly increased its expression in *Adipoq^-/-^* mice (Figure 5C). In WT mice, niacin treatment had no effect on *Arg1* expression, while in HFD-fed *Adipoq*^-/-^ mice, niacin significantly increased *Arg1* gene expression. This genotypic effect was particularly striking as HFD-fed niacin-treated *Adipoq*^-/-^ mice had a ~9-fold increase in *Arg1* gene expression compared to WT HFD-fed niacin-treated mice. Neither HFD nor niacin altered *Mrc1* expression in either genotype (Figure 5D). Interestingly, *Adipoq^-/-^* mice fed an HFD had significantly lower *Mrc1* expression than WT HFD-fed mice.

### 3.5. Effects of HFD and Niacin on Markers of Systemic Inflammation

Neither HFD nor niacin had any effect on serum concentrations of IL-6 or TNFα in either genotype (Figure 6A,B). However, it should be noted that the LFD-fed mice treated with niacin had undetectable amounts of IL-6 and TNFα in their serum. In WT mice, serum MCP-1 concentrations were not affected by diet or niacin treatment (Figure 6C). However, HFD significantly increased serum MCP-1 concentrations in *Adipoq^-/-^* mice, suggesting that the lack of adiponectin predisposes mice to an HFD-induced increase in circulating MCP-1.

## 4. Discussion

There is increasing evidence that the beneficial effects of niacin are independent of its effects on dyslipidemia [27,28,29,30,33,38]. A growing number of studies demonstrate that niacin has the ability to inhibit inflammation in a variety of tissues including adipose, colon, and brain [28,45,46]. However, the mechanisms underlying niacin’s anti-inflammatory effects are not completely understood. Niacin also increases serum and tissue concentrations of the anti-inflammatory adipokine adiponectin, through activation of the HCA_2_ receptor on adipocytes [28,40,42,47,48]. Thus, our objective was to determine if the anti-inflammatory properties of niacin were due, at least in part, to its ability to increase local and circulating adiponectin concentrations.

Obesity is associated with decreased tissue and serum adiponectin concentrations [17,49]. Consistent with previous work in our lab and others, we observed an HFD-induced decrease in adipose tissue adiponectin gene and protein expression, but no change in serum adiponectin concentrations [28]. The lack of changes observed in serum adiponectin concentrations in these mice may reflect the short time frame (11 weeks) that the mice were on a HFD. The time at which HFD-fed mice develop decreased serum or plasma adiponectin concentrations varies in the literature. Some studies, using a range of diets that contain anywhere from 40% to 60% lard, demonstrate decreased adiponectin concentrations by as early as 4 weeks on the diet [50,51,52]. However, another study similar to ours did not observe an HFD-mediated decrease in adiponectin after 10 weeks on an HFD that consisted of 60% lard [53]. While we did observe decreased tissue adiponectin levels, it is likely that we would have to increase the duration of the HFD feeding in order to obtain statistically significant changes in the serum. It is also possible that there are changes in the multimeric forms of adiponectin. However, previous work from our lab found that there was no effect on serum concentrations of high-molecular-weight adiponectin with either HFD feeding or niacin treatment [28].

Previous work from our lab and others demonstrated niacin’s ability to increase tissue and circulating adiponectin [28,47,48]. In this study, niacin increased adiponectin protein expression in HFD-fed mice. The mechanism by which niacin increases adiponectin is unclear. However, studies in our lab demonstrated that niacin-mediated increases in adiponectin are independent of changes in the gene expression of key transcription factors (PPARγ C/EBPα or SREBP-1c) and ER chaperones (ERp44, Erol-Lα and DsbA-L) known to positively regulate adiponectin gene transcription [28].

As expected, the HFD significantly impacted all of the metabolic parameters measured, including increasing body weight, EWAT weight, serum glucose concentrations, and HOMA-IR, but niacin had no effect on any of these parameters. Interestingly, the HFD-induced increase in serum insulin concentrations was more pronounced in *Adipoq^-/-^* mice. Niacin increases insulin resistance, specifically in skeletal muscle, and studies have shown that this resistance is blunted by serum adiponectin concentrations [39]. Therefore, it is not surprising that *Adipoq^-/-^* mice would have increased insulin concentrations and HOMA-IR compared to WT counterparts. Further studies are needed to investigate the metabolic phenotype of WT and *Adipoq^-/-^* mice treated with niacin.

There was a significant increase in gene expression for markers of obesity-associated adipose tissue inflammation in WT and *Adipoq^-/-^* mice fed an HFD. The general macrophage marker CD68 is a transmembrane glycoprotein expressed on monocytes and macrophages [54]. MCP-1, the protein encoded by the gene *Ccl2*, is a potent chemokine involved in the trafficking of monocytes and macrophages to adipose tissue during the development of adipose tissue inflammation [55]. Increased MCP-1 also contributes to the development of insulin resistance [56,57,58]. A few studies have also demonstrated niacin’s ability to decrease chemoattractant-mediated macrophage migration in vitro [27,59], which suggests that niacin treatment might decrease ATM content. Consistent with previous in vivo studies, we observed a significant increase in *Cd68* and *Ccl2* gene expression in the EWAT of mice fed an HFD [28]. Niacin did not alter *Cd68* gene expression, but attenuated the HFD-induced increase in *Ccl2* expression in *Adipoq^-/-^* mice. These findings suggest that niacin inhibits mechanisms of macrophage recruitment to EWAT of mice fed an HFD, but does not appear to alter overall macrophage number. Histopathologic evaluation of EWAT tissue samples further supports this conclusion, as there was a significant increase in the number of CLS in mice fed the HFD. However, niacin had no effect on CLS number in any group of mice. As MCP-1 is also secreted by adipocytes, resident ATMs, and endothelial cells [13,60], it would be interesting to determine if the changes in MCP-1 are associated with a specific cell type. The similar effect of HFD and niacin in both genotypes suggests that adiponectin does not play a role in niacin-mediated attenuation of *Ccl2* expression in adipose tissue, at least in EWAT. These findings are important, as previous work has shown that these niacin-mediated effects on *Ccl2* are dependent on activation of the HCA_2_ receptor, which also plays a role in adiponectin secretion [28]. Interestingly, *Adipoq^-/-^* mice appear to have a milder phenotype with fewer CLS than WT mice when placed on an HFD. This study is limited to the evaluation of changes in inflammation of the EWAT, a gonadal adipose tissue depot. Previous studies have shown a distinct difference in gonadal adipose tissue lipolysis and inflammation between male and female mice, with HFD feeding and changes in lipid metabolites driving gonadal adipose tissue inflammation in males, but remodeling in females. Future studies should compare the effects of niacin and adiponectin of various tissue stores, including inguinal (a subcutaneous depot), omental, and perirenal, as well as distinct differences between males and females.

Macrophage polarity is defined by patterns of cell surface receptor expression, transcription factor activation, and cytokine secretion. Adipose tissue from lean individuals contains relatively low numbers of M2 (or alternatively activated) macrophages, and with the development of obesity, there is an increase in ATMs and a phenotypic switch to M1 (or classically activated) macrophages. While most studies describe polarity with a clear dichotomy of either M1 or M2, many studies suggest that macrophage activation is on a spectrum and varies significantly based on the stimulation and microenvironment [61,62,63]. In our study, HFD increased the expression of genes typically considered as M1 macrophage markers. For example, HFD increased *Itgax* expression in WT mice and *Tnf* expression in *Adipoq^-/-^* mice. Additionally, HFD increased the expression of *Arg1*, which is typically considered an M2 macrophage marker in *Adipoq^-/-^* mice. These data support the presence of general adipose tissue inflammation.

One of the most surprising findings of our study was the opposing effects niacin had on markers of inflammation in EWAT of *Adipoq^-/-^* mice compared to WT mice. In WT mice, niacin either had no effect on or tended to reduce the expression of *Itgax, Arg1, Tnf,* and *Mrc1.* However, niacin significantly increased the expression of *Itgax* and *Arg1* in *Adipoq^-/-^* mice. In the case of *Arg1*, this increase was significantly greater than even HFD feeding alone. This finding is confounded by the overall fewer numbers of CLS in the EWAT of *Adipoq^-/-^* mice compared to WT mice. In addition, the number of CLS was not significantly increased in *Adipoq^-/-^* mice by HFD feeding and the development of obesity. Given that adipocyte inflammation is critical for normal adipose tissue remodeling [64], it is possible that in the absence of adiponectin, adipose tissue from obese individuals has fewer, but more active macrophages to allow for appropriate remodeling. Together, these findings suggest that treatment with niacin in the absence of adiponectin may promote rather than abrogate generalized ATM activation, and that adiponectin is necessary for niacin-mediated alterations in macrophage activation. In this study, adiponectin tissue concentrations were attenuated in WT mice through HFD feeding. It would be interesting to evaluate the effects of niacin administration to mice that had attenuated circulating levels of adiponectin, such as in heterozygous *Adipoq^+/-^* mice, that may more accurately reflect the low, but not absent levels of adiponectin that are observed in people with obesity. This would provide a unique and additional approach to control the levels of adiponectin in order to evaluate the specific effects of niacin under various conditions. Further studies to evaluate protein concentration and cytokine release are needed to confirm the changes in ATM polarity. In addition, it would be interesting to determine if niacin had a similar effect on circulating monocytes and immune cells in *Adipoq^-/-^* mice. Our findings are consistent with previous work in rodent models of obesity [28,29,65], and suggest that in vivo, niacin does not drive specific macrophage polarization but rather alters generalized inflammation.

Numerous studies have demonstrated that HFD feeding and obesity contribute to increases in circulating cytokines and acute phase proteins such as CRP, TNFα, and IL-6 [8,11,12,14]. The role of circulating cytokines in obesity is controversial, as they are considered both damaging and protective, and cytokines such as IL-6 regulate adipose tissue macrophage polarization [66]. We wanted to determine if the mice in our study exhibit similar evidence of systemic inflammation with the development of obesity. Our findings suggest that in WT mice, HFD-induced inflammation is localized to the tissues, with no changes in circulating inflammatory cytokines. Based on the lack of significance noted in our serum adiponectin concentrations, these findings are not surprising, and may be associated with the short timeframe of HFD feeding implemented in this study. Interestingly, a lack of adiponectin has a discordant effect on local adipose tissue production of *Ccl2* and systemic MCP-1 concentrations. This leads us to believe that the increased serum MCP-1 in *Adipoq^-/-^* mice is associated with changes in inflammation other than in adipose tissue and requires further investigation. One possible consideration is the effect of HFD-induced steatosis in the liver of the *Adipoq^-/-^* mice. When fed an HFD, *Adipoq^-/-^* mice are susceptible to the development of steatohepatitis [67], and adiponectin plays a role in attenuating liver fibrosis by inducing nitric oxide [68]. A recent paper suggests that niacin inhibits oxidative stress and lipid accumulation in hepatocytes [69]. Further studies are warranted to elucidate other potential causes for the increased serum MCP-1 concentrations in HFD-fed *Adipoq^-/-^* mice. Niacin has also been shown to inhibit monocyte adhesion to human endothelial cells [33]. Based on these findings, it would be interesting to determine if the discordant MCP-1 changes observed in the tissues and in circulation of *Adipoq^-/-^* mice manifest as a change in circulating monocyte numbers. 

## 5. Conclusions

In conclusion, our data suggest that niacin may decrease HFD-induced ATM recruitment in an adiponectin-independent manner. The mechanism behind these changes is likely directly associated with activation of the HCA_2_ rather than with changes in adiponectin. The absence of adiponectin alters niacin’s effects on some markers of inflammation in the adipose tissue but does not hinder its ability to reduce markers of macrophage recruitment.

## Figures and Tables

**Figure 1 nutrients-12-02427-f001:**
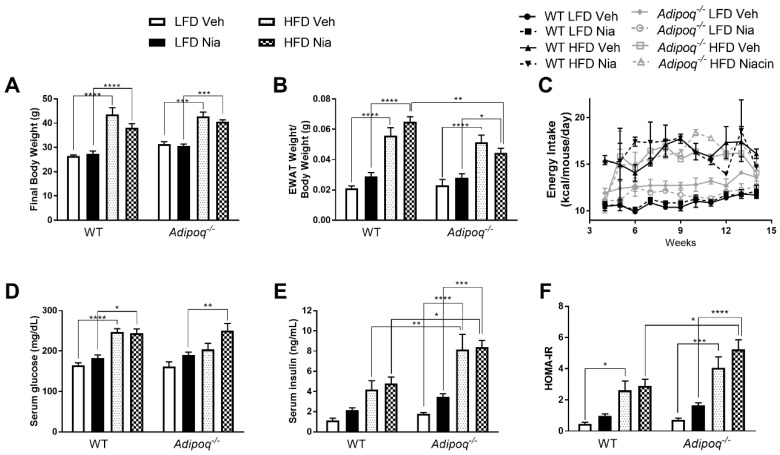
Effects of high-fat diet (HFD) and niacin on metabolic parameters. Effects of HFD and niacin on body weight (**A**), EWAT weight (**B**), energy intake (**C**), serum glucose concentrations (**D**), serum insulin concentrations (**E**), and HOMA-IR (**F**) in WT and *Adipoq^-/-^* mice. Data are presented as mean + SEM. * *p* ≤ 0.05; ** *p* ≤ 0.01; *** *p* ≤ 0.001; **** *p* ≤ 0.0001; WT low-fat diet (LFD) Veh, *n* = 8; WT LFD Nia, *n* = 8; WT HFD Veh, *n*= 8; WT HFD Nia, *n* =7; *Adipoq^-/-^* LFD Veh, *n* = 6; *Adipoq^-/-^* LFD Nia, *n* = 7; *Adipoq^-/-^* HFD Veh, *n* = 8; *Adipoq^-/-^* HFD Nia, *n* = 8.

**Figure 2 nutrients-12-02427-f002:**
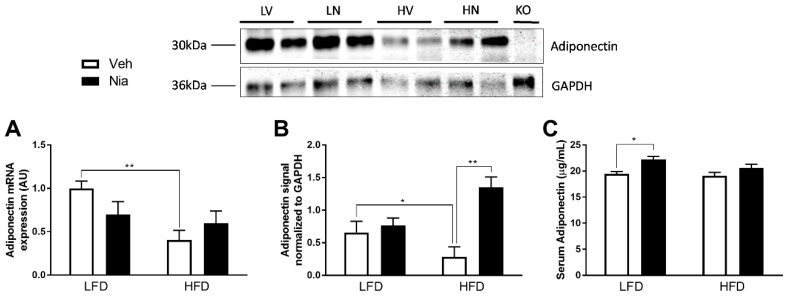
Effects of HFD and niacin on tissue and serum adiponectin concentrations. EWAT adiponectin mRNA expression (**A**), EWAT adiponectin protein expression (**B**), and serum adiponectin concentrations (**C**) in wild-type mice. Adiponectin mRNA values were normalized to *Rppl0* expression. Data are presented as mean + SEM. AU: arbitrary units. * *p* ≤ 0.05; ** *p* ≤ 0.01; LV: LFD vehicle; LN: LFD niacin; HV: HFD vehicle; HN: HFD niacin; KO: *Adipoq*^-/-^. WT LFD Veh, *n* = 8; WT LFD Nia, *n* = 8; WT HFD Veh, *n*= 8; WT HFD Nia, *n* =7.

**Figure 3 nutrients-12-02427-f003:**
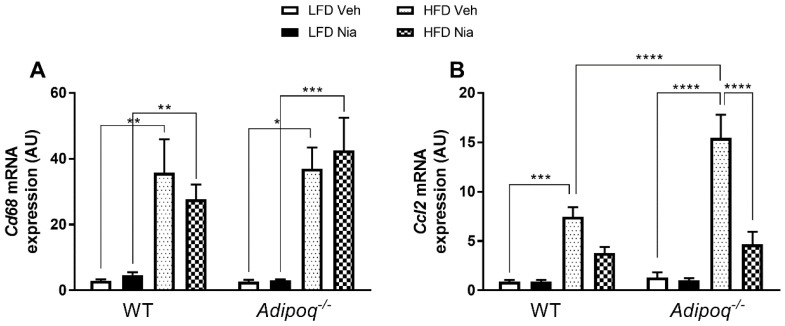
Effects of HFD and niacin on the gene expression of markers of macrophage number and migration in EWAT. Relative gene expression of the general macrophage marker *Cd68* (**A**) and the macrophage chemokine *Ccl2* (**B**) in wild-type and *Adipoq^-/-^* mice; all mRNA values were normalized to *Rppl0* expression. Data are presented as mean + SEM. AU: arbitrary units. * *p* ≤ 0.05; ** *p* ≤ 0.01; *** *p* ≤ 0.001; **** *p* ≤ 0.0001; WT LFD Veh, *n* = 8; WT LFD Nia, *n* = 8; WT HFD Veh, *n* = 8; WT HFD Nia, *n* = 7; *Adipoq^-/-^* LFD Veh, *n* = 6; *Adipoq^-/-^* LFD Nia, *n* = 7; *Adipoq^-/-^* HFD Veh, *n* = 8; *Adipoq^-/-^* HFD Nia, *n* = 8.

**Figure 4 nutrients-12-02427-f004:**
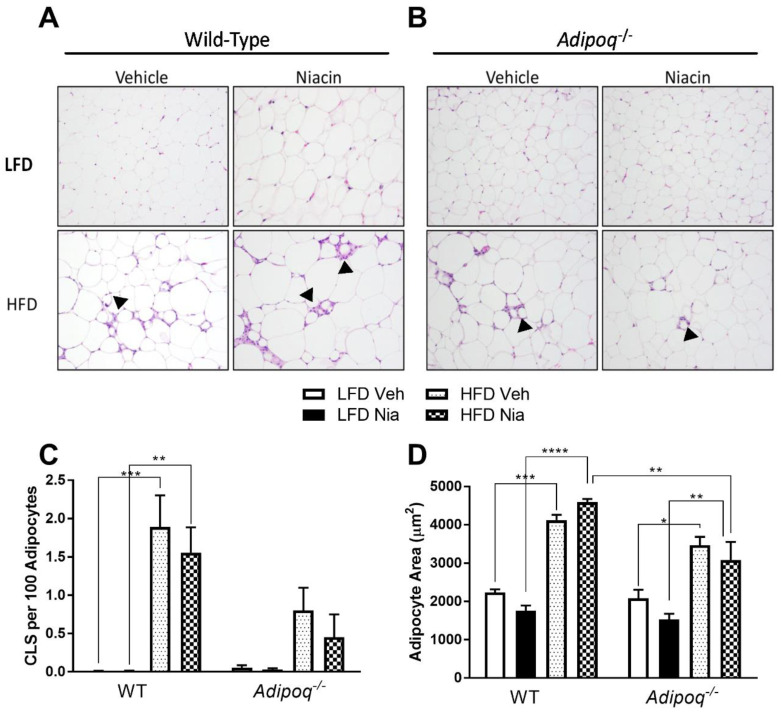
Effects of HFD and niacin on the development of crown-like structures (CLS) in EWAT. Images of hematoxylin and eosin (H&E)-stained sections of EWAT depicting CLS density for (**A**) wild-type and (**B**) *Adipoq^-/-^* mice. Number of CLS per 100 adipocytes (**C**). Mean adipocyte area (**D**). Arrowheads indicate CLS. Data are presented as mean + SEM. * *p* ≤ 0.05; ** *p* ≤ 0.01; *** *p* ≤ 0.001; **** *p* ≤ 0.0001; WT LFD Veh, *n* = 4; WT LFD Nia, *n* = 3; WT HFD Veh, *n* = 3; WT HFD Nia, *n* = 3; *Adipoq^-/-^* LFD Veh, *n* = 3; *Adipoq^-/-^* LFD Nia, *n* = 3; *Adipoq^-/-^* HFD Veh, *n* = 4; *Adipoq^-/-^* HFD Nia, *n* = 4.

**Figure 5 nutrients-12-02427-f005:**
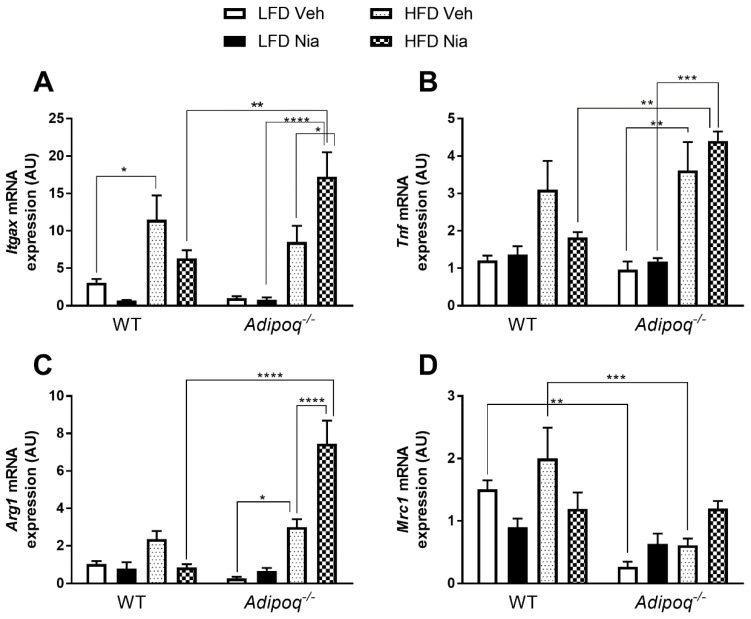
Effects of HFD and niacin on the gene expression of markers of adipose tissue macrophage polarity in EWAT. Relative gene expression of the M1 macrophage markers *Itgax* (**A**) and *Tnf* (**B**), and the M2 macrophage markers *Arg1* (**C**), and *Mrc1* (**D**) in wild-type and Adipoq^-/-^ mice. All mRNA values were normalized to Rppl0 expression. Data are presented as mean + SEM. AU: arbitrary units. * *p* ≤ 0.05; ** *p* ≤ 0.01; *** *p* ≤ 0.001; **** *p* ≤ 0.0001; WT LFD Veh, *n* = 8; WT LFD Nia, *n* = 8; WT HFD Veh, *n* = 8; WT HFD Nia, *n* = 7; *Adipoq^-/-^* LFD Veh, *n* = 6; *Adipoq^-/-^* LFD Nia, *n* = 7; *Adipoq^-/-^* HFD Veh, *n* = 8; *Adipoq^-/-^* HFD Nia, *n* = 8.

**Figure 6 nutrients-12-02427-f006:**
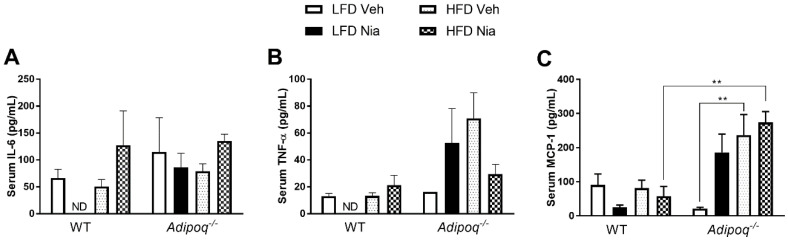
Effects of HFD and niacin on systemic markers of inflammation. Serum concentrations of the proinflammatory cytokines IL-6 (**A**) and TNF-α (**B**), and the macrophage chemokine MCP-1 (**C**), in wild-type and *Adipoq^-/-^* mice. Data are presented as mean + SEM. ND: not detectable. ** *p* ≤ 0.01; WT LFD Veh, *n* = 8; WT LFD Nia, *n* = 8; WT HFD Veh, *n* = 8; WT HFD Nia, *n* = 7; *Adipoq^-/-^* LFD Veh, *n* = 6; *Adipoq^-/-^* LFD Nia, *n* = 7; *Adipoq^-/-^* HFD Veh, *n* = 8; *Adipoq^-/-^* HFD Nia, *n* = 8.

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
