# Peer review of "The Absence of Adiponectin Alters Niacin’s Effects on Adipose Tissue Inflammation in Mice"

_nutrients, 2020, doi:10.3390/nu12082427_

Round 1

Reviewer 1 Report

Strengths:

Well written manuscript.

Straightforward study design.

Conclusion is supported by results.

Weaknesses:

Lack of data from heterozygous mice.

Obesity is associated with accumulation of visceral adiposity, analysis of EWAT may not provide the complete picture about inflammation.

Role of serum IL^ is not adequately discussed.

Authors stated that leptin and adiponectin levels determine the obesity, however they did not report serum leptin levels.

Minor comments:

Authors need to provide statistical significance associated with letter a,b...

Please provide number of animals per group (N=?) as a part of legend.

Authors need to show arrowhead towards crown like structures on Figure 4.

Author Response

Dear Reviewer,

We appreciate the time and consideration you have taken to review our manuscript carefully.  Below is a list of changes and points we have addressed regarding each of your concerns.

  • We agree that using heterozygous mice would strengthen the study. However, it is beyond the scope of this study. We have addressed this limitation in the Discussion section of the manuscript.
  • We also agree with the reviewer that adipose tissue is dynamic and can have variable effects in different depots. In future studies we would consider including other adipose tissue depots such as inguinal (a subcutaneous depot), omental and perirenal. However, these tissues are not available for evaluation in this study. We have addressed this limitation in the Discussion section of the manuscript
  • The reviewer stated, “Role of serum IL^ is not adequately discussed.” We are assuming they meant IL-6. We agree with the reviewer and have expanded the discussion accordingly.
  • While we appreciate the importance of Leptin as a circulating adipokine, we did not state that leptin and adiponectin levels determine obesity. The mention of leptin in the introduction of the manuscript was to indicate that adipokine dysregulation is a hallmark or correlate of obesity. The objective of this study was focused on the role of adiponectin. Leptin was not measured, as it was not relevant to this study.
  • We have amended the statistical analysis to provide more details about significance.
  • We have added the number of animals per group as part of the legend.
  • We have added arrowheads pointing out crown-like structures in figure 4.

Thank you again your review and suggestions have strengthened our work.

Reviewer 2 Report

Emily C. Graff and colleagues presented a study on the possible role of adiponectin in mediating the anti-inflammatory effects of niacin on adipose tissue. The used as animal model, male C57BL/6J (WT) mice and adiponectin null (Adipoq-18 /-) mice that were maintained on a low-fat diet or high-fat diet and administered with niacin or vehicle.

The novelty of this study involves the experiments conducted on adiponectin KO mice and in the findings that the lack of adiponectin alters the effects of niacin on markers of adipose tissue inflammation in high-fat-diet mice suggesting a role of this adipokine in niacin anti-inflammatory behaviour. However, the results are not always clearly described. Indeed, the authors presented in the manuscript some experiments that have been also reported in the previously paper published in PloS one 2013, 8 (8), e71285, but without clearly explaining it.

For these reasons, I suggest the manuscript to do a major revision.

In details:

  1. In all figures, in the histograms, the statistical differences are showed with different letters within each genotype. In my opinion this mode of representation is unclear. I cannot understand where there are statistical differences. The author should use * and graphical bar. Moreover, the differences between WT and adiponectin KO should also be highlighted.
  2. Figure 2. The effects of niacin on adiponectin gene expression and protein levels in WT mice (control diet versus fat diet) have already been published in PloS one 2013, 8 (8), e71285, figure 1 and figure. The authors should better explain it in the manuscript.

Author Response

Dear Reviewer.

We appreciate the time and consideration you have taken to review our manuscript carefully.  Below is a list of changes and points we have addressed regarding each of your concerns.

  • We agree with the reviewer, we believe we can provide clarity on statistical differences and have revised the figures to use asterisks and graphical bars to indicate significant differences and to evaluate the differences between genotype.
  • We wanted to provide some clarification. The mice used in this study were a separate cohort and were not involved or included in the 2013 PLOS ONE publication. In the Introduction section of this manuscript, we indicated that previous work from our lab and others has shown that niacin increases adiponectin, and we referenced the PLOS ONE paper. Since the studies in this manuscript stemmed from our findings reported in the PLOS ONE paper, we felt it was important to demonstrate niacin’s effects on adiponectin in this cohort of mice. In fact, it would be a major weakness of this study if we did not show adiponectin levels in these mice. The data reported herein are from unique WT mice and Adipoq-/- mice and have not been published elsewhere.

Thank you again your review and suggestions have strengthened our work.

Reviewer 3 Report

The manuscript be Graff et al uses a strong in vivomouse model, the adiponectin null mice, to examine known effects of niacin. The studies show that some of the effects of niacin are dependent on adiponectin while others are not. Overall, the data show niacin effects on Arg1 are adiponectin dependent, particularly during high fat feeding.

Comments for consideration:

The limitation of only using male mice should be included in discussion. Potential sex specific difference should be considered.

The figure legends should contain more information, age of mice, time of HFD. n=?

Western blot should contain molecular weight labeling.

Author Response

Dear Reviewer.

We appreciate the time and consideration you have taken to review our manuscript carefully.  Below is a list of changes and points we have addressed regarding each of your concerns.

  • We agree that only including males is a limitation of the study. We have addressed this limitation and potential sex differences in the Discussion section.
  • We have amended the legends to include the number of mice in each cohort. The age of the mice at the time of study and the initiation and duration of HFD feeding was consistent throughout the study and is provided in the methods.
  • We have added the molecular weight to the western blot images.

Thank you again your review and suggestions have strengthened our work.

Round 2

Reviewer 2 Report

Emily C. Graff and co-authors have deeply revised the Manuscript ID: nutrients-877625 entitled “The absence of adiponectin alters niacin’s effects on 3 adipose tissue inflammation in mice” addressing all the reviewer's comments. This revised manuscript is suitable for the publication in Nutrients journal.

Author Response

Dear Reviewer,

We appreciate the time and effort it takes to complete a detailed review of a manuscript and we believe your comments made our paper stronger.

Thank you for your careful review.

Emily